# Multi-model machine learning for automated identification of rice diseases using leaf image data

Rovin Tiwari[1]*, Jaideep Patel[2], Nikhat Raza Khan[3], Ajay Dadhich[4], Jay Kumar Jain[5], Kapil Gupta[6]*

1 Amity University, Gwalior, Madhya Pradesh, India, 2 Department of CSE, RGPV, Bhopal, Madhya Pradesh, India, 3 Department of Computer Science and Engineering, Oriental College of Technology, Bhopal, Madhya Pradesh, India, 4 Department of Electronics Instrumentation and Control Engineering, Engineering College Ajmer, Ajmer, Rajasthan, India, 5 Department of Mathematics, Bioinformatics and Computer Applications, Maulana Azad National Institute of Technology, Bhopal, Madhya Pradesh, India, 6 School of Computer Sciences, UPES, Dehradun, Uttarakhand, India

☯ These authors also contributed equally to this work.
* rovintiwari@gmail.com; kapil.gupta@ddn.upes.ac.in

**Data availability statement:** The leaf images utilized in our research were gathered from

## Abstract

Rice, a staple meal for about half of the world's population, is critical to global food security, especially in Asia. However, diseases have a severe impact on rice production, resulting in significant yield losses or outright crop failure. Traditional techniques of identifying rice diseases are time-consuming, labor-intensive, and rely heavily on specialist knowledge. As a result, a rapid, cost-effective, and automated method for detecting rice illnesses is critical for modernizing agricultural techniques and ensuring sustainable food production. This paper presents a novel hybrid deep-learning and machine-learning framework for automatically identifying rice plant diseases from leaf photos. We extracted deep features from rice leaf images using pre-trained CNN models—MobileNetV2, DarkNet19, and ResNet18. These features are then classified using machine learning classifiers with various kernel functions, which apply a strong 10-fold cross-validation technique to assure model reliability. Using a medium Gaussian kernel of the SVM classifier, the proposed system achieved a classification accuracy of 98.61%, specificity of 98.85%, and sensitivity of 97.25%. The framework is computationally efficient and scalable, allowing for greater dataset testing. The proposed technique provides a dependable and efficient solution for accurate identification of rice leaf diseases, reducing farmers' reliance on manual inspection and supporting timely intervention.

## Introduction

The foundation of the Indian economy is agriculture. In India, farming supports close to 70% of the population [1]. The rural populations of India are reliant on agriculture. A vital source of income for more than 58% of rural residents is farming [2]. Rice cultivation is extremely common in Asian countries, where it is a staple of agricultural and nutritional activities. Rice

various reputable sources, including the UCI and Kaggle datasets, along with specific images obtained from the IEEE dataset repository. The links to the dataset are: 1. kaggle. https://www.kaggle.com/datasets/vbookshelf/rice-leaf-diseases. 2. UCI Machine Learning Repository. https://doi.org/10.24432/C5R013. 3. IEEE Dataport. https://ieee-dataport.org/documents/indian-rice-disease-dataset-irdd (doi: 10.21227/4rmf-gd63).

**Funding:** The author(s) received no specific funding for this work.

**Competing interests:** No.

cultivation takes up a substantial percentage of fertile land in Asia, and millions of farmers work to produce it. The crop's resilience to a variety of temperatures, from tropical to temperate, helps to explain its extensive cultivation across the continent. Rice is a critical component of food security in many Asian countries, providing millions of people with their primary supply of calories and nutrients. It is the foundation of traditional diets and cultural cuisines, representing nourishment and prosperity [3]. Rice is one of the most important crops in India, but various rice diseases kill roughly 10% to 15% of the rice harvest [3]. Farmers have a long-standing agricultural difficulty in rice cultivation: efficiently managing and reducing illnesses that harm rice plants. Despite the crucial need of good crop care, many farmers lack the necessary expertise and resources to properly identify and manage diseases such as leaf blasts, brown spots, and bacterial blight. These diseases have a severe influence on rice plant growth and productivity, threatening food security and livelihoods. Early detection of infections is critical to minimizing the harm caused by these diseases, but it takes time, skill, and ongoing plant monitoring. Farmers frequently rely on manual observation, but it can be unreliable, time-consuming, and impractical, especially on large-scale farms. As a result, there is a critical need for disease diagnosis and management technologies that are both accessible and efficient to support sustainable rice production practices and strengthen farmers' resilience to agricultural difficulties [4].

However, most farmers have limited knowledge on how to protect rice plants from diseases such as leaf blast, brown spot, and bacterial blight [4]. These diseases can affect rice plants at any stage of growth, and early diagnosis is crucial to prevent significant damage. Identifying and classifying these diseases traditionally requires expertise and constant monitoring [5]. Most farmers rely on manual observation, but this method is not only time-consuming but also less reliable when monitoring large fields [6].

To improve early disease detection and classification accuracy, an automated solution is needed. This study presents a hybrid deep-learning system that uses image-based artificial intelligence tools to assess the health of rice plants. Symptoms of diseases often appear on leaves, stems, or fruits, and by analyzing leaf images with deep learning techniques, our model can identify unhealthy plant conditions. Specifically, our approach focuses on detecting three common rice diseases: bacterial blight, brown spot, and leaf blast.

Brown spot is a fungal disease caused by Bipolaris oryzae that leads to brown spots on leaves and panicles, which darken and develop a yellowish halo over time [7]. Bacterial blight, caused by Xanthomonas, results in water-soaked lesions that turn brown and yellow as the disease progresses [8]. Leaf blast, caused by Pyricularia oryzae, forms round-to-elliptical lesions with a reddish-brown ring and grey center, often leading to severe crop loss [9].

These diseases thrive in areas with high humidity and warm temperatures, causing significant yield loss if left untreated. While cultural practices like using disease-resistant varieties and maintaining proper field hygiene can help, chemical treatments such as antibiotics and copper-based fungicides may be less effective due to resistant strains.

To address these challenges, our system provides an automated, efficient approach for diagnosing these diseases from leaf images, offering detailed reports that include symptoms and management suggestions. This reduces the need for farmers to travel long distances for diagnosis, making early disease detection more accessible and effective.

## 1 Previous work

Various techniques have been proposed for detecting plant diseases using leaf images. Singh et al. [10] used image processing methods, including segmentation and classification algorithms, to detect diseases in crops such as potatoes, tomatoes, and mangoes. A genetic algorithm

was used for segmentation and classifiers such as artificial neural networks (ANN), fuzzy logic, and Bayes classifiers were explored for disease identification. The study demonstrated significant improvements in recognition accuracy with hybrid algorithms.

Khirade and Patil [11] presented an extensive review of plant disease identification using image processing, covering multiple feature extraction methods, segmentation techniques such as Otsu thresholding and k-means clustering, and classification methods, including ANN, Backpropagation Neural Networks (BPNN), and Support Vector Machines (SVM). These approaches were shown to provide high precision in the diagnosis of plant diseases.

Mahalakshmi and Srinivas [12] used the Firebird V robot for vision-based plant leaf disease detection. Their study highlighted the importance of incorporating image processing and neural networks to enhance disease detection and classification accuracy, emphasizing the need for effective and fast systems to diagnose plant diseases in real-time.

Kumar [13] reviewed several image pre-processing and segmentation techniques, including contrast enhancement, noise filtration, and morphological operators. The review also covered various classification techniques, such as K-Nearest Neighbors (KNN), Radial Basis Function, and ANN, all of which are essential for developing high-accuracy disease detection systems.

Rath and Meher [14] focused on the detection of rice blast and brown spot diseases using image processing and Radial Basis Function Neural Networks (RBFNN). Their system achieved reliable results, demonstrating the effectiveness of RBFNN in rice disease detection with a data set that contained 100 images for training and testing.

The work by Zhou et al. [15] explored a combined Faster R-CNN and FCM-KM method for the detection of rice disease. This study addressed background noise using weighted multilevel median filters and employed the R-CNN algorithm for disease classification. Their method showed improved speed and accuracy, making it suitable for real-time applications using IoT devices.

Liang et al. [16] applied a deep CNN model to identify rice blast disease, achieving promising results with a dataset comprising 2902 healthy rice leaf images and 2906 infected rice leaf images. Their study showed that CNN outperforms traditional methods, making it a strong candidate for real-world deployment.

In this work, we propose an innovative approach for automating the detection of rice plant diseases. Deep features are extracted from leaf images using pre-trained CNNs, specifically, MobileNetV2, DarkNet19, and ResNet18. These models are chosen for their robustness in feature extraction. The extracted features are merged to create a comprehensive representation, and machine learning classifiers, including ensemble classifiers with various kernel functions, k-Nearest Neighbors (KNN), and Support Vector Machine (SVM), are employed to classify the images and identify infected plants. This method promises to reduce manual labor in disease diagnosis by automating the detection process and potentially minimizing the need for manual inspection by agricultural experts.

The key contributions of this study are as follows:

1. We propose a new hybrid deep-machine learning approach that is intended to quickly and precisely identify rice illnesses. Using the best features of both paradigms, this novel method combines CNN with conventional machine-learning techniques to improve the accuracy and efficiency of rice plant disease identification.

2. The method uses a CNN to automatically identify subtle patterns linked to different diseases by extracting complex information from photos of rice plants. To complete the final classification, these deep features are then input into machine learning algorithms

like ensemble classifiers, KNN, and SVM. The process of identifying diseases is made comprehensive and reliable by the synergistic blending of deep learning and conventional machine learning approaches.

3. Our hybrid approach seeks to enable fast and accurate identification of rice diseases by combining the interpretability and generalization qualities of classical machine learning methods with the image representation capabilities of CNNs. This method takes advantage of the well-established machine learning concepts for efficient illness classification decision-making, in addition to harnessing the power of deep learning for feature extraction.

4. Produce improved classification performance that outperforms the traditional methods.

The remaining section of this paper is organized as follows: Sect 2 includes the materials and methods; Sect 3 offers the results of the simulation; Sect 4 shows how this work is discussed with previous studies; and Sect 5 concludes and shows where this study will go in the future.

## 2 Materials and methods

The presented study describes the outcomes of a real-world implementation of an artificial intelligence (AI) system for detecting rice diseases using leaf images. A publicly available dataset with a 10-FCV scheme is used to validate the presented approach. The transfer learning method is employed to re-train the utilized deep learning models. Deep features are extracted using three benchmark CNNs. Deep features of all three CNNs are merged and classified using three machine learning algorithms with different kernel functions. Fig 1 depicts the schematic view of the presented system.

### 2.1 Dataset

The leaf images used in this study were compiled from multiple reputable sources, including public repositories such as UCI, Kaggle, and the IEEE dataset repository. These sources provided a diverse collection of rice leaf images encompassing both healthy and diseased samples. To ensure consistency and quality across these heterogeneous datasets, a comprehensive curation and preprocessing workflow was implemented.

All images were converted to a uniform format (JPEG) and resized to a standard resolution of 224×224 pixels to ensure compatibility with the deep learning models. Quality control measures involved removing duplicate images, filtering out low-resolution or blurred samples,

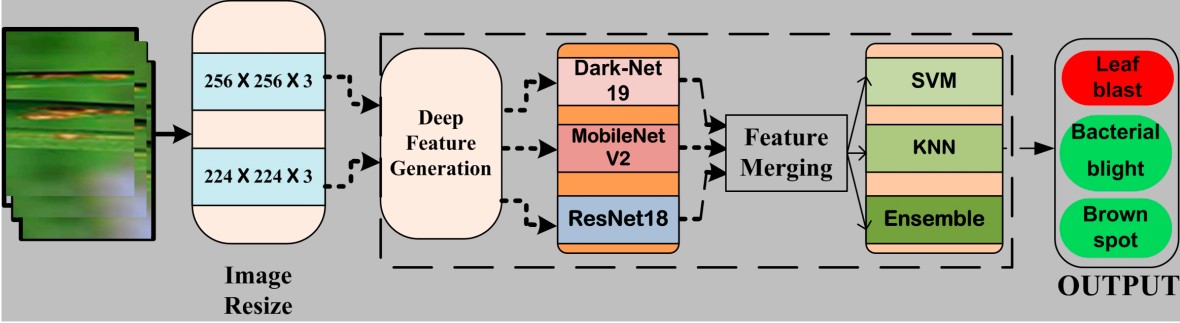

**Fig 1. Schematic view of the presented system.**

and excluding images with inconsistent lighting, backgrounds, or visible noise. Histogram analysis and edge detection techniques were also applied to verify the visual clarity and integrity of the leaf structures.

To prevent class imbalance and ensure robust model training, we carefully maintained a balanced distribution across the three targeted rice disease categories: bacterial blight, leaf blast, and brown spot. The final curated dataset consists of 1,030 images of rice spot, 1,130 images of rice blast, and 1,536 images of rice blight. Fig 2 illustrates six representative images from the dataset, each depicting a distinct disease class.

**2.1.1 Verify the image clarity.** After images are collected, their quality is evaluated. When the input image quality satisfies the predetermined standards, noise reduction preprocessing is started, and the input matrix of the model is put through quality assessment. When an image is rejected by the model and a corresponding message is displayed, it means that the image quality is not up to par. To assess the quality of input images, CNN uses the BRISQUE algorithm that is built into the 'Image-quality' package. CNN libraries use the scores that the BRISQUE algorithm produces for each input image to evaluate the quality of the image. Through transformative feature computations, this approach, which uses image pixels to compute features, is highly effective in assessing the quality of photographs of rice plants.

Data Labeling Procedure: The dataset for this investigation consists of tagged images acquired from publicly available sources and agricultural research sites. Each image was pre-labeled with the appropriate disease class based on the metadata provided by the data sources. To guarantee consistency, we carefully checked the labels to assure their accuracy.

## 2.2 Convolutional Neural Network (CNN)

CNN-based frameworks are widely studied in current research, especially for various classification challenges. CNN's attraction is its capacity to substitute an automated system for conventional, laborious approaches, including feature evaluation, selection, and classification.

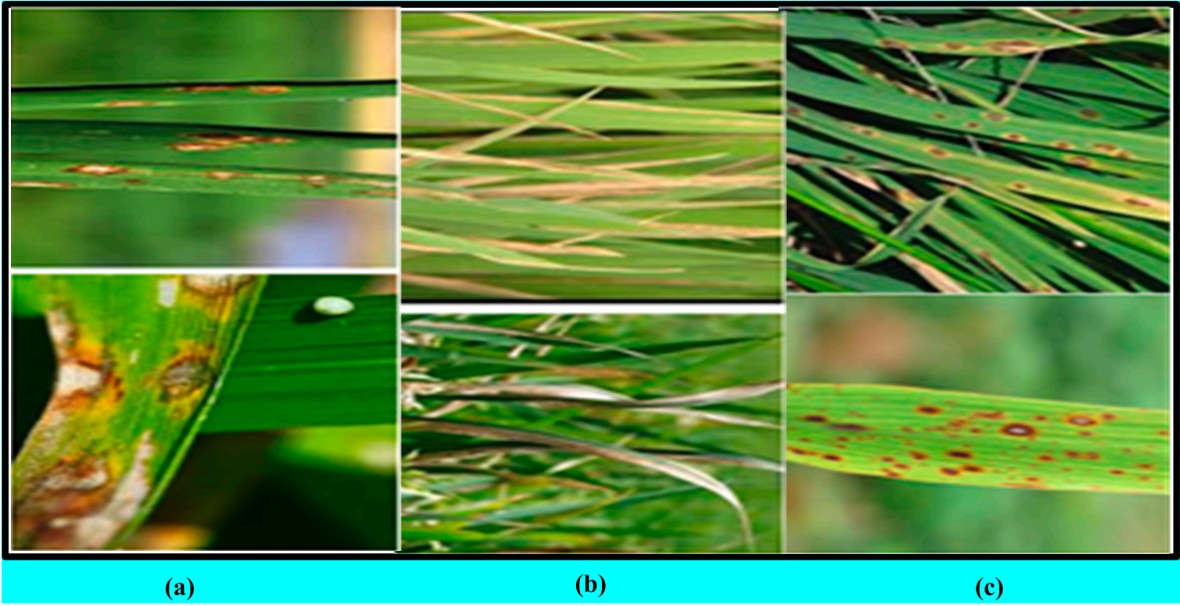

**Fig 2. Rice-leaf images of (a) Brown Spot Leaf, (b) Bacterial Blight, and (c) Blast Disease.**

The process is streamlined and made more flexible by this automated method. CNN is capable of performing a wide range of pattern recognition tasks, including object localization, segmentation, detection, and image classification. Because of its adaptability, CNN-based frameworks have been widely used and investigated in a variety of disciplines [17].

Deep learning approaches are used to diagnose rice leaf illnesses. These approaches include deep feature extraction, transfer learning to fine-tune benchmark Convolutional Neural Networks (CNNs), and extensive training and validation of the proposed framework. All of these techniques work together to make the model more successful in precisely identifying and categorizing different rice plant illnesses. Furthermore, the use of deep learning has demonstrated its effectiveness in a variety of biomedical tasks, such as segmentation issues, diagnosis of various diseases, and signal and image categorization. The adaptability shown in these applications highlights the potential of deep learning techniques, which makes them useful instruments for dealing with difficult problems in a variety of fields, such as plant pathology and agriculture [18]. One type of representation-based learning (RBL) that has a structured design is called deep learning. An input layer, several convolutional layers (Conv2DL), pooling layers (PL), a few fully connected layers (FCL), and an output classification layer are the usual components of this architecture. Together, the essential elements of this architecture allow the model to automatically extract hierarchical representations from the input data; this makes deep learning very useful for intricate tasks like image categorization and recognition. Convolution operations are carried out by the Conv2DL layers to capture spatial patterns; dimensionality is reduced by the pooling layers; and the fully connected layers allow the model to establish complex correlations between features for precise classification in the output layer. Deep learning stands out as a potent paradigm in many domains, such as pattern recognition and computer vision, thanks to its hierarchical and automated representation learning [18].

In this research, three benchmarks CNNs, DarkNet19, ResNet18, and MobileNetV2, are used for disease detection using leaf images.

A sophisticated CNN model created especially for object detection is called Darknet-19. It is a crucial part of the object detection models created by Joseph Redmon and Ali Farhadi as part of their YOLO (You Only Look Once) series. When it comes to real-time object detection tasks, the YOLO models are renowned for their effectiveness and quickness [19,20]. With 19 convolutional layers, Darknet-19 is a lighter variant of the YOLO series models, which makes it appropriate for situations when resources are limited. Darknet-19 retains a high degree of accuracy in object detection tasks even with its decreased complexity. The architecture of the model utilizes several convolutional layers to extract features, which facilitates the analysis and identification of objects in a picture. With models like Darknet-19, the YOLO framework has been widely applied in a variety of fields, such as robotics, autonomous vehicles, and surveillance, where precise and real-time object detection is essential. Google has created a small CNN model called MobileNetV2, which is intended for effective and efficient object recognition and classification in embedded and mobile applications. It is the replacement for the initial MobileNet model, which was created to offer a thin CNN model that could function well on portable devices. Because MobileNetV2 is built on an inverted residual structure, it can achieve good accuracy at low computational complexity and small model sizes. It can carry out image classification and object recognition tasks because it was trained on the ImageNet dataset, which is a sizable collection of images annotated with various item types. MobileNetV2's effective use of computing resources is one of its primary characteristics, which makes it a good choice for deployment on mobile and embedded devices with constrained memory and processing power. It is ideal for applications like video surveillance and autonomous cars since it can operate in real-time on these devices. An input

image with dimensions of $224 \times 224$ is used by MobileNetV2. There are two more features in MobileNetV2. The first kind of bottlenecks are linear ones. The quickest route between bottlenecks is the second. MobileNetV2 can be used for object detection, feature generation, and classification. A vast collection of images annotated with various item types makes up the ImageNet dataset, which is used to train the 18-layer CNN known as ResNet18. By using an image as input and producing a set of class probabilities for each object it identifies in the image, it may carry out tasks related to object identification and image classification. Microsoft Research created the ResNet18 model, which is intended for use in object identification and picture classification applications. This model is a variation of ResNet, which was created to help with the difficulty of training extremely deep CNNs. The $224 \times 224$ input image is accepted by the network.

## 2.3 Feature fusion strategy

To enhance the robustness and discrimination ability of our model, we utilized three pre-trained CNN models—MobileNetV2, DarkNet19, and ResNet18—to extract deep features from the input leaf images. Each model outputs a one-dimensional feature vector after the global average pooling layer. Let:

- $F_1 \in \mathbb{R}^{n_1}$ denote the feature vector extracted from MobileNetV2,
- $F_2 \in \mathbb{R}^{n_2}$ denote the feature vector extracted from DarkNet19,
- $F_3 \in \mathbb{R}^{n_3}$ denote the feature vector extracted from ResNet18.

These feature vectors are concatenated along the feature axis to form a comprehensive fused feature vector:

$$F_{\text{fused}} = \left[ F_1 \parallel F_2 \parallel F_3 \right] \in \mathbb{R}^{n_1 + n_2 + n_3} \tag{1}$$

This fused vector $F_{\text{fused}}$ is then fed into traditional machine learning classifiers such as Support Vector Machine (SVM), K-Nearest Neighbors (KNN), and ensemble learning models to perform the final classification. The concatenation strategy leverages the complementary feature representations learned by the different CNN architectures, thereby improving the model's overall performance.

## 2.4 Classification of features using traditional machine learning methods

The potential of learned image features for rice disease identification is being researched in addition to the application of the transfer learned network for screening [21]. The network activations from the various learned model layers are collected and fed into the SVM, KNN, and Ensemble-proven state-of-the-art classifiers. In this paper, a 10-FCV approach is used to construct the model, which helps to minimize potential bias during model generation. The efficacy of the suggested feature extraction model is assessed using the classification results obtained. Brief details for SVM, KNN, and Ensemble classifiers are summarized as follows: SVM is a form of supervised machine learning method that can be used for classification, regression, and outlier detection. They work by identifying a hyperplane in a high-dimensional space that isolates dissimilar classes as much as possible [22]. SVMs excel at managing high-dimensional data and complex decision boundaries in the context of classification, displaying robustness to overfitting and making them acceptable for jobs with small or moderate-sized datasets. To improve SVM's flexibility in capturing varied patterns, this work employs five kernel functions: linear, cubic, quadratic, coarse Gaussian, and

medium Gaussian. K-Nearest Neighbors (KNN) is a simple and effective method for classification and regression tasks that uses the instance-based learning principle. It saves training data and predicts future instances based on their similarity to previously saved data [23,24]. KNN, which is frequently used for nonparametric classification, selects the ideal $k$ (number of neighbors) value and assigns labels based on the most common labels from the $k$-nearest neighbors. While KNN is simple and flexible, its computational complexity can be a disadvantage for large datasets. To increase the versatility of the KNN approach, five distinct kernel functions are used in this study: Cosine, Cubic, Fine, Weighted, and Medium. When compared to a single estimator, ensemble approaches integrate predictions from numerous base estimators to improve generalizability and robustness [25]. The goal is to improve model generalization by decreasing overfitting and increasing prediction stability. This research employs four ensemble methods, including Bagging, Subspace, RUSBoosted, and Subspace, to leverage the collective strength of various base models for superior task performance.

## 2.5 Performance metrics

The developed hybrid system achieves several key performance metrics to assess its effectiveness. These metrics include accuracy ($A_{CC}$), sensitivity ($S_{EN}$), specificity ($S_{PE}$), F-1 measure, precision ($PRC$), negative prediction value ($N_{PV}$), and area under the curve ($AUC$). Each of these metrics provides valuable insights into different aspects of the system's performance, such as its ability to correctly classify both positive and negative instances, its precision in identifying true positives, and its overall discriminatory power. By considering multiple performance characteristics, the proposed approach aims to comprehensively evaluate its superiority and effectiveness in automated rice plant disease detection. The mathematical representation of these performance characteristics is given as [26,27]:

$$A_{CC}(\%) = \frac{T_{Pos} + T_{Neg}}{T_{Pos} + F_{Neg} + T_{Neg} + F_{Pos}} \times 100 \tag{2}$$

$$S_{EN}(\%) = \frac{T_{Pos}}{T_{Pos} + F_{Neg}} \times 100 \tag{3}$$

$$S_{PE}(\%) = \frac{T_{Neg}}{T_{Neg} + F_{Pos}} \times 100 \tag{4}$$

$$F-1\ score = \frac{2 \times T_{Pos}}{2 \times T_{Pos} + F_{Pos} + F_{Neg}} \tag{5}$$

$$PRC = \frac{T_{Pos}}{T_{Pos} + F_{Pos}} \tag{6}$$

$$N_{PV} = \frac{T_{Neg}}{T_{Neg} + F_{Neg}} \tag{7}$$

where, true positive ($T_{Pos}$), false positive ($F_{Pos}$), true negative ($T_{Neg}$), and false negative ($F_{Neg}$) are the confusion matrix parameters.

## 3 Simulation setup and findings

The experiment is run on a single CPU with a Core i7 processor, 24-GB RAM, a 1-TB SSD, and a 512-GB hard disk, using the MATLAB platform. To augment the dataset and enhance its size, we employed a combination of common data augmentation techniques. These included rotation, reflection, zooming, and shearing of the input images (randomly rotated from $-5^0$ to $5^0$, zoom range (0.8 to 1.2), width and height shift (range = 0.2), and shear transformation (shear range = 0.2 for both horizontal and vertical directions). Rotation involved rotating the images by various angles, while reflection encompassed flipping the images horizontally and vertically. Additionally, shearing was applied to introduce geometric transformations by distorting the images along their axes. By systematically applying these augmentation methods, we effectively diversified the dataset, enriching it with variations of the original images to enhance model robustness and generalization capabilities. A 10-fold cross-validation (10-FCV) scheme is used for a more robust evaluation. This approach involves dividing the extracted features into ten equal parts, with one part reserved for testing, one for validation, and the remaining eight for model training. The process is repeated ten times, each time using a different partition for testing and validation while utilizing the remaining partitions for training. This strategy ensures comprehensive evaluation and validation of the model's performance across multiple iterations, enhancing its reliability and generalization capability. The Adam optimizer is used with a starting learning rate of 0.0001 to maximize the learning rate and reduce cross-entropy loss. Table 1 shows the hyperparameter settings for the second and third studies. This comprehensive setup ensures that the proposed framework is thoroughly evaluated, taking into account both hardware specifications and robust cross-validation processes. Fig 3 shows the training progress graphs yielded for (A) MobileNetV2 (B) Resnet-18 (C) DarkNet19. Table 2 gives the overall classification accuracy yielded by classifying features extracted by individual CNNs and merged features of all CNNs, using various machine learning algorithms. It is evident from Table 2 highest classification accuracy of 98.9% is achieved by classifying merged features of all CNNs by applying the SVM algorithm with a medium Gaussian kernel function. Various performance metrics are evaluated, to

**Table 1. Details of hyperparameters used the secondond and third experiments.**

| Classifier | Kernel Function | Box Constraint Level | Kernel Scale | Multi-class Method | Standardize Data |
|---|---|---|---|---|---|
| SVM | Linear | 1 | Automatic | One-vs-One | True |
| | Cubic | | | | |
| | Quadratic | | | | |
| | Coarse Gaussian | | | | |
| | Medium Gaussian | | | | |
| **Classifier** | **Kernel Function** | **Number of Neighbors** | **Distance Metric** | **Distance Weight** | **Standardize Data** |
| KNN | Cosine | 10 | Cosine | Equal | True |
| | Cubic | 10 | Minkowiski | Equal | |
| | Fine | 1 | Euclidean | Equal | |
| | Weighted | 10 | Euclidean | Squared Inverse | |
| | Medium | 10 | Euclidean | Equal | |
| **Classifier** | **Ensemble Method** | **Number of Split** | **Number of Learner** | **Learning Rate** | **Subspace Dimension** |
| Ensemble | Bag | 20 | 30 | 0.1 | 1 |
| | Subspace Discriminant | | | | |
| | RUSBoosted | | | | |
| | Subspace KNN | | | | |

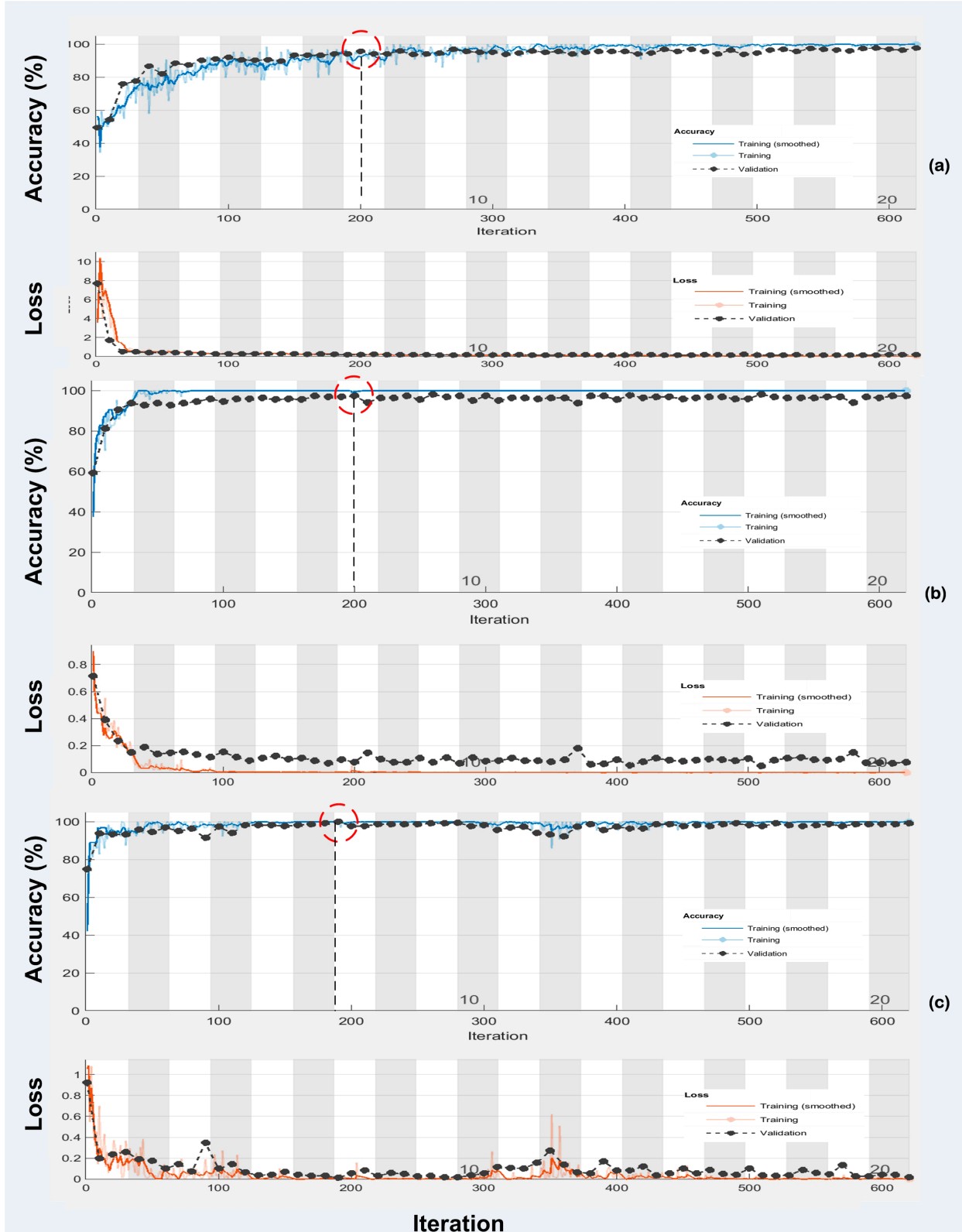

**Fig 3. Training progress graphs yielded for: (A) MobileNetV2 (B) Resnet-18 (C) DarkNet19.**

**Table 2**. Summary of classification accuracies (%) obtained using classifiers.

| Classifier | Kernel Function | Mobile-NetV2 Features | Dark-Net19 Features | Res-Net18 Features | Merged Features |
|---|---|---|---|---|---|
| | | ($A_{CC}$%) | | | |
| SVM | Linear | 94.6 | 93.5 | 92.1 | 95.2 |
| | Quadratic | 94.6 | 93.8 | 95.7 | 96.2 |
| | Cubic | 94.2 | 93.7 | 91.1 | 97.6 |
| | Medium Gaussian | 94.3 | 95.5 | 96.3 | **98.9** |
| | Coarse Gaussian | 91.6 | 92.1 | 93.8 | 96.9 |
| KNN | Fine | 92.6 | 91.1 | 90.7 | 93.6 |
| | Medium | 92.8 | 90.6 | 91.2 | 94.4 |
| | Cosine | 89.4 | 87.8 | 88.5 | 90.2 |
| | Cubic | 86.9 | 87.2 | 88.1 | 92.8 |
| | Weighted | 88.9 | 89.5 | 92.1 | 93.0 |
| Ensemble | Bag | 90.3 | 93.3 | 93.6 | 95.5 |
| | Subspace Discriminant | 89.8 | 91.4 | 91.9 | 93.3 |
| | Subspace KNN | 92.5 | 93.4 | 92.7 | 93.9 |
| | RUSBoosted | 93.2 | 96.5 | 96.5 | 97.7 |

check the effectiveness of the presented SVM algorithm with medium Gaussian kernel function to classify merged features of all CNNs into different disease classes. Fig 4 shows the bar chart of the obtained classification accuracy using different kernels of the machine learning classifier.

Table 3 gives the overall performance metrics obtained for the SVM algorithm with a medium Gaussian kernel to classify merged features.

It can be noted from Table 3, that the SVM classifier with a medium Gaussian kernel has achieved the $S_{EN}$ of 97.25%, $S_{PE}$ of 98.85%, F1-score of 97.2, PRC of 98.6, and $N_{PV}$ of 97.8 in classifying merged features of all three explored CNNs.

Fig 5 represents the overall receiver operating characteristic (ROC) plots for medium Gaussian SVM.

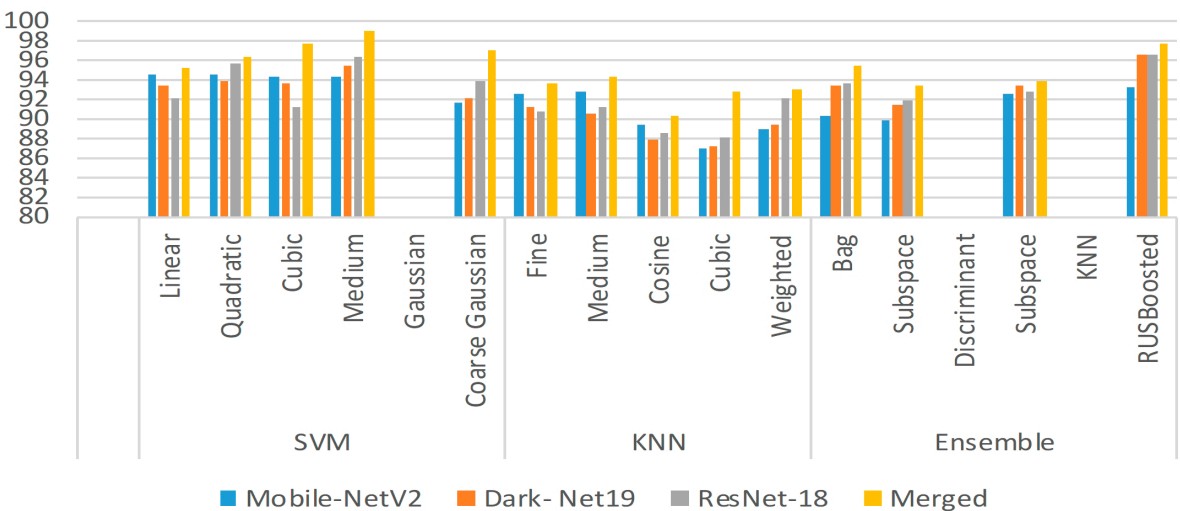

**Fig 4. Bar chart of the obtained classification accuracy.**

**Table 3**. Performance metrics obtained for SVM classifier with medium Gaussian kernel.

| Parameters | Performance (in %) |
|---|---|
| $A_{CC}$ | 98.61 |
| $S_{EN}$ | 97.25 |
| $S_{PE}$ | 98.85 |
| $F1$-score | 97.2 |
| $PRC$ | 98.6 |
| $N_{PV}$ | 97.8 |
| $AUC$ | 0.98 |

Fig 5 specifies that the classifier obtains a value of 0.98 for True Positive Rate (TPR) vs. false Positive fraction (FPR).

## 4 Discussion

Plant disease diagnosis and classification from leaf images has emerged as a fascinating area of disease detection at the nexus of computer science and agriculture. In agriculture, a variety of computer vision/artificial intelligence-based methods are utilized to identify and categorize illnesses from images of plants. Farmers encounter difficulties in recognizing infections due to their limited awareness of disease symptoms, which can lead to crop losses. Manual detection is a laborious and imprecise method in which farmers observe plants at every stage of growth. To solve these issues in the context of rice plants, this study makes use of image processing and computer vision, particularly deep hybrid approaches. Unlike existing approaches, our method capitalizes on the complementary strengths of CNNs for image feature extraction and conventional machine learning algorithms for robust classification. By leveraging

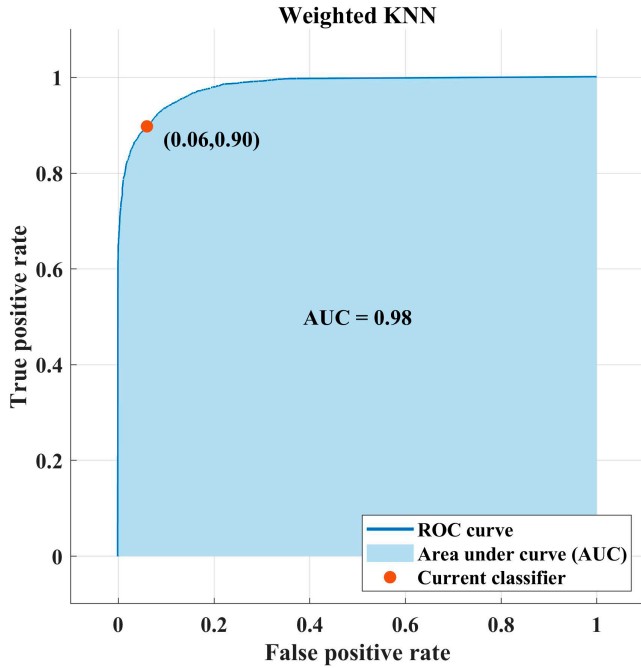

**Fig 5. ROC plots achieved for SVM classifier with medium Gaussian kernel.**

CNNs to automatically discern intricate patterns associated with various diseases from rice leaf images, we pave the way for more nuanced and accurate classification outcomes. Subsequently, these extracted deep features are seamlessly integrated into ensemble classifiers, KNN, and SVM algorithms, enabling comprehensive disease detection with enhanced reliability. The main objective of this research is to use hybrid deep machine learning to detect three major rice plant illnesses to maximize rice output in terms of both quality and quantity. The main goal is to help rice farmers, analysts, and other stakeholders by offering a precise and timely method for identifying rice plant illnesses. The two stages of the suggested model are feature extraction and classification. During the feature extraction phase, a predetermined collection of pictures of rice plant leaves—including those with bacterial leaf blight, brown spot disease, and leaf blast disease—is used to extract deep features from pre-trained deep learning models. The diagnosis, classification, and cure recommendation processes are under the purview of the disease prediction phase using machine learning models. Based on information obtained from agricultural specialists, the algorithm classifies the type of disease and proposes a specific therapy in addition to predicting whether a leaf is healthy or sick. A summary of the comparison of rice plant disease detection using the AI-based technique is presented in Table 4.

The proposed hybrid deep-learning model was specifically designed to address the three primary goals outlined in the research:

1. Determine whether a rice leaf is healthy or not: The model successfully achieved this goal by accurately classifying rice leaf images as either healthy or diseased, thereby enabling farmers to identify potential health issues affecting their crops.
2. Identify the sort of disease: Through comprehensive analysis and classification, the model accurately identified various types of rice diseases, including bacterial blight, leaf blasts, and brown spots, among others. This achievement fulfills the second goal of the research, providing farmers with crucial information about the specific diseases affecting their crops.
3. Make a recommendation for how to treat the illness that is anticipated: While the manuscript may not explicitly detail treatment recommendations, the successful identification and classification of rice diseases empower farmers to take timely and informed action to address the identified illnesses.

The proposed technology offers several potential benefits for farmers, particularly in the context of rice plant disease management:

1. The proposed system provides a reliable and efficient method for diagnosing and classifying plant diseases from leaf images, leveraging advancements in computer vision and artificial intelligence.

**Table 4. Comparison of rice plant disease detection using the AI-based technique.**

| S.No. | Authors | Proposed technique | Performance (in %) |
|---|---|---|---|
| 1. | Krishnamoorthy et al. [28] | InceptionResNetV2 | ACC=95.67 |
| 2. | Jiang et al. [29] | ResNet50 CNN model | ACC=98.0 |
| 3. | Coulibalya et al. [30] | VGG16 model | ACC=95.0 |
| 4. | Proposed Method | Hybrid model | ACC=98.6 |

2. By automating the disease detection process, farmers can overcome challenges associated with limited awareness of disease symptoms, thereby reducing the risk of crop losses and improving overall yield.

3. This study offers a promising solution to enhance disease surveillance and management practices in rice cultivation, ultimately empowering farmers with tools to make informed decisions and optimize crop health and productivity.

## 5 Conclusion and future directions

Early detection and treatment of plant diseases are critical for reducing crop output losses. Our suggested hybrid model for rice disease prediction showed a considerable improvement in disease detection, with classification accuracy of 98.6%, specificity of 98.85%, and sensitivity of 97.25%. These findings demonstrate that the model can accurately identify the three major rice plant diseases, allowing for timely intervention and treatment. Our solution, which automates the disease detection process, assists farmers in swiftly diagnosing and treating rice problems, decreasing their reliance on labor-intensive manual procedures. This not only saves time and money, but also boosts agricultural output by enabling for early disease intervention. The model's ability to classify diseases based on leaf photos gives farmers with a simple and inexpensive tool for managing crop health more effectively. While existing image processing techniques have proved beneficial, they frequently lack the comprehensive analysis and speed required for prompt response in the field. Our method bridges these gaps by merging CNN-based feature extraction with machine learning classifiers, providing a more reliable and efficient disease diagnosis solution. Furthermore, this approach is versatile and may be expanded to cover other crops, increasing its application in a variety of agricultural settings. As technology progresses and user needs change, the suggested framework can be expanded to fulfill the demand for more efficient, scalable, and user-friendly agricultural disease management solutions.

## Author contributions

**Conceptualization:** Rovin Tiwari.

**Formal analysis:** Jaideep Patel.

**Methodology:** Nikhat Raza Khan.

**Software:** Ajay Dadhich.

**Supervision:** Kapil Gupta.

**Validation:** Jay Kumar Jain, Kapil Gupta.

**Visualization:** Kapil Gupta.

**Writing – review & editing:** Kapil Gupta.

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
