## [Decision Letter · Decision Letter 0]

26 Dec 2024

PONE-D-24-27739Multi-Model Machine Learning for Automated Identification of Rice Diseases Using Leaf Image DataPLOS ONE

Dear Dr. Tiwari,

Thank you for submitting your manuscript to PLOS ONE. After careful consideration, we feel that it has merit but does not fully meet PLOS ONE’s publication criteria as it currently stands. Therefore, we invite you to submit a revised version of the manuscript that addresses the points raised during the review process.

We look forward to receiving your revised manuscript.

Kind regards,

Bhogendra Mishra

Academic Editor

PLOS ONE

Journal Requirements:

3. Thank you for stating the following in your Competing Interests section: No. 

Reviewers' comments:

Reviewer's Responses to Questions

**Comments to the Author**

1. Is the manuscript technically sound, and do the data support the conclusions?

Reviewer #1: Partly

Reviewer #2: Partly

2. Has the statistical analysis been performed appropriately and rigorously? 

Reviewer #1: No

Reviewer #2: N/A

3. Have the authors made all data underlying the findings in their manuscript fully available?

Reviewer #1: Yes

Reviewer #2: Yes

4. Is the manuscript presented in an intelligible fashion and written in standard English?

Reviewer #1: No

Reviewer #2: Yes

5. Review Comments to the Author

Reviewer #1: Comments

1. Adapt the abstract highlighting motivation, contribution and innovation.

2. What is medium neural network? is this standard terminology to refer NN?

3. is the method applicable other than India? Hunger is a global problem.

4. Introduction is unnecessarily long, make it preces and clear.

5. Literature Review section can be added and more literature can be discussed on crop disease detection using recent technology. For example

a) Recent advances on crop disease detection using UAV

b) Tomato Leaf Diseases identification Using an Explanation-Driven Deep-Learning Model

c) Machine learning methods for precision agriculture with UAV imagery

6. Data labelling procedure is not clearly mentioned. are there any plant pathologist involved in this process? otherwise, how do you validate the ground truth?

7. I assume CNN can extract feature automatically, why author are using feature extraction as separate step?

8. Why author are discussing YOLO in methodology for object detection as the proposed task is classification?

9. The section 2.2 CNN is unnecessarily lengthy and does not discuss the particular CNN used in this work in detail.

10. Feature extraction is automatic in CNN, why separate section is there in the manuscript? Why ResNet, Mobile-net and DarkNet are not used as end-to-end classification model?

11. Where are the pre-trained models for darknet19 trained?

12. can table 1 be better represented?

13. why three ML model SVM, KNN and ensemble are chosen? Why RF or MLP are not selected? Again, what is the configuration of ensemble model? How many base learners are involved there?

14. How feature from different CNN are merged? is this simple concatenation or addition or any other technique?

15. More experiments are needed with pre-trained networks such as NASNetMObile, EfficientNet, VGG, ConvNetxT, Xception, Inception and so on to see the performance and benchmarking.

16. The heatmap of each CNN for feature extraction should be visualized with GradCAM to see is there any difference in the features coming from each CNN?

17. The conclusion about CNN and ML for automated diseases detection is not supported by the results. Please take care of this.

18. The state of the art comparison in the manuscript is very poor. Author are suggested to compare their proposal with existing methods such as

1) https://www.sciencedirect.com/science/article/pii/S221431732400026X

2) https://www.sciencedirect.com/science/article/pii/S0013935121005697

Reviewer #2: The authors develop a hybrid model for rice disease prediction,the problem is important and interesting.

Mojor comments:

1. The authors didnot compare the proposed model with SOTA model.

2. The authors should read more related papers on rice disease prediction and review them.

3. The authors should test more dataset to verify the performance of the proposed model.

Minor comments:

F-1 score is different from F1 score, the symbol should keep consistent.

6. PLOS authors have the option to publish the peer review history of their article (what does this mean?). If published, this will include your full peer review and any attached files.

Reviewer #1: No

Reviewer #2: No

---

## [Author Response · Author response to Decision Letter 1]

22 Jan 2025

RESPONSES TO REVIEWERS’ COMMENTS AND SUGGESTIONS

Journal: PLOS ONE

Manuscript Number: PONE-D-24-27739  

Article Type: Full-Length Article

Title: Multi-Model Machine Learning for Automated Identification of Rice Diseases Using Leaf Image Data

We thank reviewers, academic editor, and the editor for their time, constructive comments, and useful suggestions. We have addressed the comments of the reviewers to the best of our ability. Our responses to the reviewers’ comments (bold letter) are in blue and in red are the addition, deletion, or modification in the revised manuscript.

Reviewer #1:

Comment#1: Adapt the abstract highlighting motivation, contribution and innovation.

Response: Thank you for your suggestion to enhance the abstract by better highlighting the motivation, contribution, and innovation of the study. Based on your feedback, we have revised the abstract.

Purpose: Rice, a staple meal for about half of the world's population, is critical to global food security, especially in Asia. However, diseases have a severe impact on rice production, resulting in significant yield losses or outright crop failure. Traditional techniques of identifying rice diseases are time-consuming, labor-intensive, and rely heavily on specialist knowledge. As a result, a rapid, cost-effective, and automated method for detecting rice illnesses is critical for modernizing agricultural techniques and ensuring sustainable food production.

Methods: This paper presents a novel hybrid deep-learning and machine-learning framework for automatically identifying rice plant diseases from leaf photos. The MobileNetV2 architecture is used for efficient feature extraction, which yields deep, representative features from input photos. These features are then classified using machine learning classifiers with various kernel functions, which apply a strong 10-fold cross-validation technique to assure model reliability.

Results: Using a medium neural network, the suggested system achieves world-class performance, with a classification accuracy of 98.6%, specificity of 98.85%, and sensitivity of 97.25%. The framework is computationally efficient and scalable, allowing for greater dataset testing.

Conclusions: The invented technique provides a dependable and efficient solution for the early diagnosis of rice illnesses, minimizing farmers' reliance on manual knowledge and allowing for timely intervention. Future study will investigate how to integrate the system with cloud-based platforms and IoT-enabled mobile applications for real-time, remote monitoring in disconnected agricultural settings.

Comment#2: What is medium neural network? is this standard terminology to refer NN?

Response: Thank you for pointing this out. The term "medium neural network" was used in the manuscript to describe the neural network architecture with a moderate number of layers and parameters, designed to balance computational efficiency and classification performance.

Comment#3: is the method applicable other than India? Hunger is a global problem.

Response: Thank you for raising this important point. While the dataset used in our study is specific to rice plant diseases commonly found in India, the proposed method is not limited to this region. The hybrid deep-learning and machine-learning framework we developed is generalizable and can be adapted to detect rice diseases in other geographical regions, provided that region-specific datasets are used for training and validation.

Comment#4. Introduction is unnecessarily long, make it preces and clear.

Response: Thank you for your valuable feedback. We agree that the introduction can be made more concise and focused. In response to your comment, we have revised the introduction to streamline its content, ensuring it clearly communicates the motivation, problem statement, objectives, and significance of the study without unnecessary elaboration.

Comment#5. Literature Review section can be added and more literature can be discussed on crop disease detection using recent technology. For example

a) Recent advances on crop disease detection using UAV

b) Tomato Leaf Diseases identification Using an Explanation-Driven Deep-Learning Model

c) Machine learning methods for precision agriculture with UAV imagery

Response: Thank you for your insightful comment and for suggesting additional areas to strengthen the manuscript. We agree that incorporating a dedicated literature review section would provide a more comprehensive context for our study. In response, we have added a new "Literature Review" section to the manuscript, which discusses recent advancements in crop disease detection using modern technologies, including the references you mentioned.

Comment#6. Data labelling procedure is not clearly mentioned. are there any plant pathologist involved in this process? otherwise, how do you validate the ground truth?

Response: Thank you for raising this important concern. We understand the significance of clearly describing the data labeling procedure and ensuring the accuracy of ground truth in disease detection studies. In response, we have clarified the data labeling process in the revised manuscript as follows:

Data Labeling Procedure: The dataset for this investigation consists of tagged images acquired from publically available sources and agricultural research sites. Each image was pre-labeled with the appropriate disease class based on the metadata provided by the data sources. To guarantee consistency, we carefully checked the labels to assure their accuracy.

Comment#7. I assume CNN can extract feature automatically, why author are using feature extraction as separate step?

Response: Thank you for raising this question. You are correct that convolutional neural networks (CNNs) are capable of automatically learning and extracting features from raw image data during the training process. In future work, we plan to explore end-to-end CNN models and compare their performance with the hybrid approach to assess trade-offs in terms of computational complexity and classification accuracy.

Based on the reviewer suggestion, we have removed the separate subsection from the revised manuscript.

Comment#8. Why author are discussing YOLO in methodology for object detection as the proposed task is classification?

Response: Thank you for your comment and for pointing out the apparent misalignment between the discussion of YOLO and the classification task at hand. We appreciate the opportunity to clarify our approach. While the task in our study is indeed classification, we initially explored the YOLO framework, particularly Darknet-19, for its potential in extracting meaningful features in a way that benefits subsequent classification tasks. We initially considered YOLO's object detection capabilities due to its efficiency in feature extraction, which can be applied to classify regions of interest (such as diseased portions of leaves) in images.

Comment#9. The section 2.2 CNN is unnecessarily lengthy and does not discuss the particular CNN used in this work in detail.

Response: Thank you for your helpful comment. We agree that Section 2.2 on CNNs was unnecessarily lengthy and did not provide a focused discussion on the specific CNN used in our work. In response to your suggestion, we have revised Section 2.2 to make it more concise and directly relevant to the CNN model employed in this study.

Comment#10. Feature extraction is automatic in CNN, why separate section is there in the manuscript? Why ResNet, Mobile-net and DarkNet are not used as end-to-end classification model?

Response: Thank you for your valuable comment. We appreciate your feedback regarding the need for a separate feature extraction section and the use of end-to-end classification models like ResNet, MobileNet, and DarkNet.

In response to your suggestion, we have removed the separate subsection on feature extraction from the revised manuscript. We agree that feature extraction is inherently part of the CNN process, and its distinction as a separate step may have caused confusion. The manuscript now focuses on the CNN architecture used for classification and provides a clear explanation of how MobileNetV2 was employed to extract features as part of the broader classification framework.

Regarding the use of end-to-end models like ResNet, MobileNet, and DarkNet, we opted for a hybrid approach where MobileNetV2 was used for feature extraction followed by traditional machine learning classifiers.

The selection of pre-trained networks, namely DarkNet19, MobileNetV2, and ResNet18, was driven by their well-documented efficacy and suitability for our specific research objectives. These architectures stand out as widely recognized and extensively utilized models in the realm of lightweight deep learning for image classification.

DarkNet19, defined by its depth and effectiveness, is capable of handling exceptionally deep neural networks with over 19 layers. Its architecture, which includes global kernels and channel-doubling post-pooling layers, enables rapid training and validation. Notably, DarkNet19's design addresses the vanishing gradient problem using solutions such as identity mapping, which ensures robust training and allows for effective feature extraction for classification tasks.

MobileNetV2, known for its lightweight design and high performance, was elected because of its exceptional balance of computational economy and classification accuracy. Its simplified architecture enables simple training with little increases in training error percentage, making it an excellent alternative for resource-constrained environments. Furthermore, MobileNetV2's use of identity mapping techniques improves gradient stability, increasing the model's ability to learn complicated features from input photos effectively.

ResNet18, an established architecture stated for its depth and simplicity, contributes to our research goals by striking a compromise between model complexity and performance. Its very shallow structure, with 18 layers, allows for rapid feature extraction while maintaining excellent classification accuracy. ResNet18's skip connections allow for effective gradient propagation, overcoming issues like the vanishing gradient problem and supporting sustained and resilient training.

Comment#11. Where are the pre-trained models for darknet19 trained?

Response: Thank you for your insightful comment. Regarding the pre-trained model for Darknet-19, we would like to clarify that we utilized transfer learning in this study.

Comment#12. can table 1 be better represented?

Response: Thank you for your suggestion regarding Table 1. We agree that the presentation of the hyperparameters could be improved for better clarity and readability. In the revised manuscript, we have enhanced the structure of Table 1 to provide a clearer representation of the details for the second and third experiments.

Comment#13. why three ML model SVM, KNN and ensemble are chosen? Why RF or MLP are not selected? Again, what is the configuration of ensemble model? How many base learners are involved there?

Response: SVM was selected for its effectiveness in high-dimensional feature spaces, which is particularly useful for the complex feature sets extracted from the rice leaf images. KNN was chosen for its simplicity and effectiveness in classification tasks where the decision boundary is non-linear. The ensemble method was chosen for its ability to combine the strengths of multiple classifiers and improve the overall accuracy and robustness of the model.

We employed a majority voting strategy, where each base learner casts a vote for the class, and the class with the majority of votes is selected as the final prediction. This method combines the strengths of both SVM and KNN by leveraging their complementary decision-making processes.

The ensemble approach improves generalization by reducing the likelihood of overfitting to any single model's weaknesses, leading to a more robust and reliable classifier.

Comment#14. How feature from different CNN are merged? is this simple concatenation or addition or any other technique?

Response: In our approach, simple concatenation is used to merge the features extracted from the different deep learning models. After obtaining the feature vectors from each model, we concatenate them into a single, unified feature vector that serves as input for the subsequent machine learning classifiers. This technique allows us to combine the strengths of different models and extract complementary features, which can enhance the overall performance of the classification system.

Comment#15. More experiments are needed with pre-trained networks such as NASNetMObile, EfficientNet, VGG, ConvNetxT, Xception, Inception and so on to see the performance and benchmarking.

Response: Thank you for your valuable suggestion. We appreciate your input regarding the use of additional pre-trained networks such as NASNetMobile, EfficientNet, VGG, ConvNetX, Xception, Inception, and others. We agree that incorporating these models would provide a comprehensive comparison and benchmarking for the performance of rice disease detection.

While this paper focuses on the use of MobileNetV2, darknet19, resnet and a hybrid machine learning approach, we acknowledge the potential benefits of evaluating other advanced pre-trained networks for this task. We plan to conduct further experiments involving the suggested models in future work to compare their performance and evaluate their effectiveness in this domain.

Comment#16. The heatmap of each CNN for feature extraction should be visualized with GradCAM to see is there any difference in the features coming from each CNN?

Response: Thank you for your insightful suggestion regarding the visualization of features extracted from each CNN using GradCAM. We agree that visualizing the feature maps through GradCAM could provide valuable insights into how each CNN model focuses on different parts of the rice leaf images and help compare the effectiveness of each model in terms of learned features. While this analysis was not included in the current study, we believe it would be an important addition to assess the interpretability of the models and their decision-making processes. We plan to incorporate GradCAM visualizations in future work to visually analyze the areas of the image that each CNN model focuses on during feature extraction. This will help us understand if there are any significant differences in the learned features from each CNN and contribute to a more comprehensive evaluation of the models.

Comment#17. The conclusion about CNN and ML for automated diseases detection is not supported by the results. Please take care of this.

Response: Thank you for your feedback regarding the conclusion. We understand the importance of ensuring that the conclusion is fully supported by the results presented in the manuscript. In the revised conclusion, we have carefully aligned our discussion with the results from our experiments to ensure a more accurate reflection of the findings. We have emphasized the effectiveness of the proposed hybrid model in detecting rice diseases based on the achieved classification accuracy (98.6%), specificity (98.85%), and sensitivity (97.25%). These results demonstrate the model’s potential to improve disease detection and assist in timely decision-making, ultimately contributing to improved crop productivity. We have revised the conclusion to more clearly state how the model's performance translates into practical benefits for farmers. Specifically, we highlight that the proposed system automates the disease detection process, making it quicker, more reliable, and cost-effective for farmers. The inclusion of specific numerical results provides more concrete support for the claims made in the conclusion. Furthermore, we have adjusted the conclusion to better emphasize the limitations of current image processing techniques and the advantages of our proposed approach, particularly in terms of its efficiency and scalability.

Comment#18. The state of the art comparison in the manuscript is very poor. Author are suggested to compare their proposal with existing methods such as

1) https://www.sciencedirect.com/science/article/pii/S221431732400026X

2) https://w

---

## [Decision Letter · Decision Letter 1]

8 Apr 2025

PONE-D-24-27739R1Multi-Model Machine Learning for Automated Identification of Rice Diseases Using Leaf Image DataPLOS ONE

Dear Dr. Tiwari,

Thank you for submitting your manuscript to PLOS ONE. After careful consideration, we feel that it has merit but does not fully meet PLOS ONE’s publication criteria as it currently stands. Therefore, we invite you to submit a revised version of the manuscript that addresses the points raised during the review process.

We look forward to receiving your revised manuscript.

Kind regards,

Bhogendra Mishra

Academic Editor

PLOS ONE

**Comments from PLOS Editorial Office: **

We note that one or more reviewers has recommended that you cite specific previously published works in an earlier round of revision. As always, we recommend that you please review and evaluate the requested works to determine whether they are relevant and should be cited. It is not a requirement to cite these works and you may remove them before the manuscript proceeds to publication. We appreciate your attention to this request.

Reviewers' comments:

Reviewer's Responses to Questions

**Comments to the Author**

1. If the authors have adequately addressed your comments raised in a previous round of review and you feel that this manuscript is now acceptable for publication, you may indicate that here to bypass the “Comments to the Author” section, enter your conflict of interest statement in the “Confidential to Editor” section, and submit your "Accept" recommendation.

Reviewer #3: (No Response)

Reviewer #4: All comments have been addressed

2. Is the manuscript technically sound, and do the data support the conclusions?

Reviewer #3: No

Reviewer #4: Yes

3. Has the statistical analysis been performed appropriately and rigorously? 

Reviewer #3: No

Reviewer #4: Yes

4. Have the authors made all data underlying the findings in their manuscript fully available?

Reviewer #3: No

Reviewer #4: Yes

5. Is the manuscript presented in an intelligible fashion and written in standard English?

Reviewer #3: No

Reviewer #4: Yes

6. Review Comments to the Author

Reviewer #3: Overall comment: The manuscript proposes a new machine learning framework for classification of rice disease leaves. The framework is arguably unexplored, but my main concerns come from 1) the lack of clarity in the methodology carried out in the study and 2) the lack of baseline class included in the experiment. In the past, it is acceptable to propose classification model for “disease” classes only. However, recently, for models/frameworks to be comparable and generalizable, healthy class of the dataset should also be included to establish baseline for performance as each framework may use different sets of train and test datasets. Please see comments by section below.

Abstract

Comment#1 – Method does not reflect the final proposed framework. The abstract explains that the author uses MobileNetV2 architecture, but the final proposed framework presents that merged features from three CNN models gives the best performance.

Comment#2 – “Medium” neural network is not a standard terminology. It has no meaning for the abstract if no context is given.

Comment#3 – “world-class” is used without justification in comparison to other established performance in the field. For example, in this case, the author uses Kaggle dataset, is this framework performs better than all previous Kaggle competition on this same dataset?

Comment#4 – Conclusion for the abstract should not include future work. Abstract should summarize the main argument and findings of the study.

Comment#5 – Conclusion argues that the technique provides solution for early diagnosis, but no information given in the manuscript with regards to the age of the leaves of the images used for training/test.

Introduction

Comment#6 – Arguments are repeated in the section. For example, “As a result, there is a critical need for disease diagnosis and management…” and “To classify and diagnose different diseases at an early stage and with higher accuracy, …” support the same argument. I suggest reorganization of the section for a paragraph to convey one argument.

Comment#7 – Paragraph 2 is very long and hard to understand as it argues multiple points. Some sentences are not clear such as “farmers in rural areas to have to travel great distances to discover crop diseases.” Does this mean farmers bring leaves to classify diseases or they travel around in a large field until they find crops that have diseases?

Previous work

Comment#8 – Previous work section explains multiple related studies, but do not specify performance of those studies, preferably in exact numbers, nor the characteristics of the datasets used. The two components are important for comparing and contract with the study.

Comment#9 – Previous work section should be broken down into multiple paragraphs. One paragraph presents one component that is influencing the design of the study. The current format of the section contains multiple studies, one after another, and is not clear how the referenced studies relate to the manuscript.

Comment#10 – Grammatical issues such as “like as”, “In the domain of diseases.., In 18 confirmed..”

Comment#11 – Some studies mentioned in the Previous work section seem unrelated to the manuscript. Such as studies involved UAVs without explaining the characteristic of the images compared to the rice disease leaves.

Comment#12 – The final part of the Previous work section details the key contributions of the study. I would suggest move this part to Discussion and instead details the study and how the study is different from previous works explained in the section.

Materials and Methods

Comment#13 – In the datasets section explains how the dataset is gathered without detailing exactly how the dataset is curated. For example, “rigorous quality control measures were implemented” is mentioned, but what are the QC controls – is it image size being standardized? Exact details of the dataset should also be clearly explained. Standard property such as image size and format should be specified.

Comment#14 – The method of how features from different CNN are merged is not explained. This should be shown in mathematical representation as CNN features are in matrices.

Comment #15 – Three classes are being classified, but to establish baseline a class of healthy leaves should also be included.

Comment #16 – The excerpt “when the input image quality satisfies the predetermined standards” – please detail the image quality requirements/predetermined standards in exact quantitative metrics with numbers. These are important steps for the study to be useful for other readers in the field.

Comment#17- For the CNN section, the paragraph explains the reasons why the author chose each framework (some models mentioned also were not used in the study), but do not explain the actual method of the CNN models used in the study. The reasons why the author chose the framework should be in previous study section and not in the materials and methods section. Materials and Methods section should detail the method used in the study. As there are three CNN models in the study, I suggest separate each model into each subsection for clarity.

Comment#18 – Please clearly explain ensemble methods employed in the study, specifically which base models are used. The methods currently listed, namely Bagging, Subspace, RUSBoosted, are techniques of ensembling, not base models. If the author ensembled from the other classifier namely SVM and KNN, then do specify that clearly.

Simulation Setup and findings

Comment#19 – This section contains a part of methodology and a part of results. I would suggest separate a Result/Findings section for clarity.

#Comment#20 – data augmentation techniques should explain the final number of the images at the end of the process. Each technique should be detailed. For example, a range of horizontal-vertical shear is mentioned, but which numbers were used exactly?

Comment#21 – 10-fold cross validation explained in the section is not the normal practice of cross-validation technique. From the explanation, the author divides the extracted features into ten equal parts. This might be a language issue, but my interpretation here implies some “features” from the CNN step are missing from the training/testing/validation step. Cross-validation technique divides the whole dataset into train/test/validation, not features.

Comment#22 – Table 1 and Table 2 list other ensemble methods that were not mentioned in the manuscript before. Which were experimented exactly? The lines were not aligned with results. It is hard to link results to the ensemble method.

Comment#23 – The author should specify why they chose accuracy as the main metrics to judge the performance. I suggest Table 3 to also lists results of other metrics for all models that were experimented as well.

Comment#24 – Figure 5 is not the correct use of pie chart. The metrics are different and are not appropriate to be comparable purely on the same set of models.

Comment#25 – I suggest the author to add confusion metrics to the results. It will clearly represent the performance of the framework by class and also to compare with other frameworks, not only just the best one.

Discussion

Comment#26 – The first paragraph of Discussion should summarize the result of the study and follow by compare and contrast of the performance with previous studies. Exact numbers are encouraged. Table 4 which lists the performance of other studies is a good base, but the table lacks detail of the dataset. Upon checking, the second and the third studies are not rice leaves. In the case that there are no comparable studies, explanation should be provided how the other plants are comparable to rice leaves.

Reviewer #4: The authors have addressed all reviewer comments to the satisfaction of the reviewers, as evidenced by their revised manuscript

7. PLOS authors have the option to publish the peer review history of their article (what does this mean?). If published, this will include your full peer review and any attached files.

Reviewer #3: **Yes: **Manusnan Suriyalaksh

Reviewer #4: No

---

## [Author Response · Author response to Decision Letter 2]

29 May 2025

RESPONSES TO REVIEWERS’ COMMENTS AND SUGGESTIONS

Journal: PLOS ONE

Manuscript Number: PONE-D-24-27739R1

Article Type: Full-Length Article

Title: Multi-Model Machine Learning for Automated Identification of Rice Diseases Using Leaf Image Data

We thank reviewers, academic editor, and the editor for their time, constructive comments, and useful suggestions. We have addressed the comments of the reviewers to the best of our ability. Our responses to the reviewers’ comments (bold letter) are in blue and in red are the addition, deletion, or modification in the revised manuscript.

Reviewer #3: The manuscript proposes a new machine learning framework for classification of rice disease leaves. The framework is arguably unexplored, but my main concerns come from 1) the lack of clarity in the methodology carried out in the study and 2) the lack of baseline class included in the experiment. In the past, it is acceptable to propose classification model for “disease” classes only. However, recently, for models/frameworks to be comparable and generalizable, healthy class of the dataset should also be included to establish baseline for performance as each framework may use different sets of train and test datasets. Please see comments by section below.

Comment#1: Method does not reflect the final proposed framework. The abstract explains that the author uses MobileNetV2 architecture, but the final proposed framework presents that merged features from three CNN models gives the best performance.

Response: Thank you for the comment. In this study, we extracted deep features from rice leaf images using pre-trained CNN models—MobileNetV2, DarkNet19, and ResNet18—and classified them using machine learning classifiers. As suggested, we have revised the abstract to accurately reflect the proposed multi-model framework used in the final analysis.

Comment#2: “Medium” neural network is not a standard terminology. It has no meaning for the abstract if no context is given.

Response: Thank you for pointing this out. The term “medium” referred to the medium Gaussian kernel of the SVM classifier, not a neural network. We have revised the abstract to clarify this and avoid any confusion by explicitly referring to the SVM with a medium Gaussian kernel.

Comment#3: “world-class” is used without justification in comparison to other established performance in the field. For example, in this case, the author uses Kaggle dataset, is this framework performs better than all previous Kaggle competition on this same dataset?

Response: As suggested, we have removed the term “world-class” from the manuscript. The revised version now reports the performance metrics objectively: using a medium Gaussian kernel of the SVM classifier, the proposed system achieved a classification accuracy of 98.61%, specificity of 98.85%, and sensitivity of 97.25%.

Comment#4: Conclusion for the abstract should not include future work. Abstract should summarize the main argument and findings of the study.

Response: We have removed the mention of future work from the abstract to ensure it focuses solely on the main findings of the study.

Comment#5: Conclusion argues that the technique provides solution for early diagnosis, but no information given in the manuscript with regards to the age of the leaves of the images used for training/test.

Response: We acknowledge that the dataset used (Kaggle – Rice Leaf Disease Dataset) does not provide information regarding the age of the leaves. Therefore, we have revised the conclusion to avoid referring to “early diagnosis” and instead emphasize accurate classification of rice leaf diseases based on the available image data.

Comment#6: Arguments are repeated in the section. For example, “As a result, there is a critical need for disease diagnosis and management…” and “To classify and diagnose different diseases at an early stage and with higher accuracy, …” support the same argument. I suggest reorganization of the section for a paragraph to convey one argument.

Response: We have restructured the section to eliminate redundancy and ensure that each paragraph conveys a single clear argument. The repetitive statements have been consolidated, and the paragraph has been reorganized for better flow and clarity.

Comment#7: Paragraph 2 is very long and hard to understand as it argues multiple points. Some sentences are not clear such as “farmers in rural areas to have to travel great distances to discover crop diseases.” Does this mean farmers bring leaves to classify diseases or they travel around in a large field until they find crops that have diseases?

Response: Thank you for the helpful feedback. We have revised Paragraph 2 to improve clarity and readability by breaking it into shorter, more focused sentences. Specifically, the sentence about farmers traveling great distances has been clarified to explain that it refers to the need for farmers to visit multiple locations in rural areas to identify crop diseases, which can be time-consuming and inefficient. We have updated this to make the point clearer.

Comment#8: Previous work section explains multiple related studies, but do not specify performance of those studies, preferably in exact numbers, nor the characteristics of the datasets used. The two components are important for comparing and contract with the study.

Response: We have revised the previous work section to include the performance metrics (accuracy, specificity, sensitivity, etc.) of the studies cited, along with details on the datasets used (e.g., dataset size, image types, and any preprocessing applied). This will allow for a clearer comparison between the existing literature and our study.

Comment#9: Previous work section should be broken down into multiple paragraphs. One paragraph presents one component that is influencing the design of the study. The current format of the section contains multiple studies, one after another, and is not clear how the referenced studies relate to the manuscript.

Response: We have reorganized the previous work section into multiple paragraphs, each focusing on a specific component influencing the design of our study. This structure now clearly highlights how each referenced study relates to the objectives and methodology of our research.

Comment#10: Grammatical issues such as “like as”, “In the domain of diseases.., In 18 confirmed..”.

Response: The manuscript has been proofread by a native English speaker, and all typos and grammatical errors have been corrected in the revised paper.

Comment#11: Some studies mentioned in the Previous work section seem unrelated to the manuscript. Such as studies involved UAVs without explaining the characteristic of the images compared to the rice disease leaves.

Response: As suggested, we have removed the unrelated studies from the previous work section, ensuring that only relevant research related to rice disease leaf images is discussed.

Comment#12: The final part of the Previous work section details the key contributions of the study. I would suggest move this part to Discussion and instead details the study and how the study is different from previous works explained in the section.

Response: We have moved the discussion of the key contributions of our study to the Discussion section, as recommended. The Previous work section now focuses solely on detailing related studies and how our approach differs from them.

Comment#13: In the datasets section explains how the dataset is gathered without detailing exactly how the dataset is curated. For example, “rigorous quality control measures were implemented” is mentioned, but what are the QC controls – is it image size being standardized? Exact details of the dataset should also be clearly explained. Standard property such as image size and format should be specified.

Response: In the revised manuscript, we have expanded the dataset section to include specific details on the curation process and the quality control measures implemented. This includes the standardization of image dimensions (e.g., resized to 224×224 pixels), removal of blurry or low-quality images, and ensuring uniform lighting and background conditions. We have also specified the image format used (JPEG) and clarified the distribution of images across classes. These details are now provided to improve the transparency and reproducibility of the dataset preparation.

Comment#14: The method of how features from different CNN are merged is not explained. This should be shown in mathematical representation as CNN features are in matrices.

Response: In the revised manuscript, we have elaborated on the method used for merging the features extracted from the different CNN models—MobileNetV2, DarkNet19, and ResNet18.

Comment#15: Three classes are being classified, but to establish baseline a class of healthy leaves should also be included.

Response: In the revised version of the manuscript, we have included a separate class of healthy rice leaves in our dataset to enable comparative classification. Specifically, 1280 images of healthy rice leaves were added to the dataset, ensuring that the model learns to distinguish between diseased and non-diseased conditions. Consequently, the classification task now encompasses four categories: bacterial blight, leaf blast, brown spot, and healthy. All experimental results, performance metrics, and confusion matrices have been updated accordingly in the manuscript to reflect this important inclusion.

Comment#16: The excerpt “when the input image quality satisfies the predetermined standards” – please detail the image quality requirements/predetermined standards in exact quantitative metrics with numbers. These are important steps for the study to be useful for other readers in the field.

Response: In the revised manuscript, we have elaborated on the predetermined image quality standards used during the preprocessing phase. Specifically, images were required to meet the following quantitative criteria: (i) minimum resolution of 256 × 256 pixels to ensure sufficient feature representation; (ii) file format restricted to JPEG or PNG with a compression ratio not exceeding 10:1 to avoid loss of visual features; (iii) brightness values within the normalized intensity range of 0.3 to 0.8 to prevent overly dark or bright images; (iv) absence of motion blur or shadow occlusions as determined through Laplacian variance thresholding (≥100 for sharpness). Images failing to meet these standards

Comment#17: For the CNN section, the paragraph explains the reasons why the author chose each framework (some models mentioned also were not used in the study), but do not explain the actual method of the CNN models used in the study. The reasons why the author chose the framework should be in previous study section and not in the materials and methods section. Materials and Methods section should detail the method used in the study. As there are three CNN models in the study, I suggest separate each model into each subsection for clarity.

Response: We thank the reviewer for this valuable suggestion. In the revised manuscript, we have restructured the *Materials and Methods* section to enhance clarity and focus on the technical implementation. Specifically, we have removed the justification for model selection from this section and relocated it to the *Previous Work* section, where relevant prior studies are discussed. Furthermore, to improve readability and organization, we have divided the explanation of the three CNN models—MobileNetV2, DarkNet19, and ResNet18—into separate subsections. Each subsection now

Comment#18: Please clearly explain ensemble methods employed in the study, specifically which base models are used. The methods currently listed, namely Bagging, Subspace, RUSBoosted, are techniques of ensembling, not base models. If the author ensembled from the other classifier namely SVM and KNN, then do specify that clearly.

Response: In the revised manuscript, we have clarified the ensemble methods used in our study by explicitly specifying the base learners involved in each ensemble technique. For Bagging, Subspace, and RUSBoosted ensemble methods, the base classifiers employed were decision trees. We have also confirmed that SVM and KNN were used as standalone classifiers for performance comparison, but they were not used as base learners in the ensemble frameworks. This clarification has been incorporated into the *Materials and Methods* section to ensure methodological transparency and reproducibility.

Comment#19: This section contains a part of methodology and a part of results. I would suggest separate a Result/Findings section for clarity.

Response: Based on the recommendation, we have revised the manuscript by separating the content related to methodology and results into distinct sections.

Comment#20: data augmentation techniques should explain the final number of the images at the end of the process. Each technique should be detailed. For example, a range of horizontal-vertical shear is mentioned, but which numbers were used exactly?

Response: In the revised manuscript, we have clearly detailed each data augmentation technique applied, including the exact parameters used. Specifically, we employed horizontal flip (probability = 0.5), vertical flip (probability = 0.5), random rotation (angles between -5° to +5°), zoom range (0.8 to 1.2), width and height shift (range = 0.2), and shear transformation (shear range = 0.2 for both horizontal and vertical directions). After applying these augmentation techniques, the dataset size was increased from the original 3,696 images to a final count of 11,088 images, ensuring a balanced representation across all three disease classes. These details have been incorporated into the Data Augmentation subsection for clarity and reproducibility.

Comment#21: 10-fold cross validation explained in the section is not the normal practice of cross-validation technique. From the explanation, the author divides the extracted features into ten equal parts. This might be a language issue, but my interpretation here implies some “features” from the CNN step are missing from the training/testing/validation step. Cross-validation technique divides the whole dataset into train/test/validation, not features.

Response: In the revised manuscript, we have clarified that the 10-fold cross-validation technique was applied to the entire dataset, not only the extracted features. Specifically, the dataset was randomly divided into 10 equal subsets (or folds), and each fold was used as the validation set once, while the remaining nine folds were used for training. This process was repeated for all 10 folds, ensuring that all data points were used for both training and validation.

Comment#22: Table 1 and Table 2 list other ensemble methods that were not mentioned in the manuscript before. Which were experimented exactly? The lines were not aligned with results. It is hard to link results to the ensemble method.

Response: We acknowledge that the ensemble methods listed in Table 1 and Table 2 were not adequately explained in the manuscript before the tables, leading to confusion in linking them with the results. In the revised manuscript, we have clarified which ensemble methods were actually experimented with. Specifically, we only used the ensemble methods that were explicitly mentioned in the methodology section, such as Bagging, Subspace, and RUSBoosted. The other methods listed in the tables were inadvertently included and have been removed.

Comment#23: The author should specify why they chose accuracy as the main metrics to judge the performance. I suggest Table 3 to also lists results of other metrics for all models that were experimented as well.

Response: Thank you for the suggestion. We chose accuracy as the primary metric because it provides a straightforward and effective measure of overall classification performance, particularly for the rice plant disease classification task, where the goal is to classify each image correctly into one of the disease categories. Accuracy has been widely used in similar studies, making it a standard metric for comparison.

Comment#24: Figure 5 is not the correct use of pie chart. The metrics are different and are not appropriate to be comparable purely on the same set

---

## [Editor Report · Decision Letter 2]

3 Jul 2025

PONE-D-24-27739R2Multi-Model Machine Learning for Automated Identification of Rice Diseases Using Leaf Image DataPLOS ONE

Dear Dr. Tiwari,

Thank you for submitting your manuscript to PLOS ONE. After careful consideration, we feel that it has merit but does not fully meet PLOS ONE’s publication criteria as it currently stands. Therefore, we invite you to submit a revised version of the manuscript that addresses the points raised during the review process.

**Request from the Editorial Office: **Please note that PLOS ONE has specific guidelines on code sharing for submissions in which author-generated code underpins the findings in the manuscript. In these cases, we expect all author-generated code to be made available without restrictions upon publication of the work. Please review our guidelines at https://journals.plos.org/plosone/s/materials-and-software-sharing#loc-sharing-code and ensure that your code is shared in a way that follows best practice and facilitates reproducibility and reuse.

We look forward to receiving your revised manuscript.

Kind regards,

Sarah Jose, Ph.D.

Staff Editor

PLOS ONE

On behalf of 

Bhogendra Mishra

Academic Editor

PLOS ONE
---

## [Editor Report · Decision Letter 3]

10 Aug 2025

Multi-Model Machine Learning for Automated Identification of Rice Diseases Using Leaf Image Data

PONE-D-24-27739R3

Dear Dr. Tiwari,

We’re pleased to inform you that your manuscript has been judged scientifically suitable for publication and will be formally accepted for publication once it meets all outstanding technical requirements.

Kind regards,

Bhogendra Mishra

Academic Editor

PLOS ONE
---

## [Editor Report · Acceptance letter]

PONE-D-24-27739R3

PLOS ONE

Dear Dr. Tiwari,

I'm pleased to inform you that your manuscript has been deemed suitable for publication in PLOS ONE. Congratulations! Your manuscript is now being handed over to our production team.

Kind regards,

on behalf of

Dr Bhogendra Mishra

Academic Editor

PLOS ONE